# Machine learning based energy-free structure predictions of molecules, transition states, and solids

Dominik Lemm[1], Guido Falk von Rudorff [1] & O. Anatole von Lilienfeld [1,2 ✉]

The computational prediction of atomistic structure is a long-standing problem in physics, chemistry, materials, and biology. Conventionally, force-fields or ab initio methods determine structure through energy minimization, which is either approximate or computationally demanding. This accuracy/cost trade-off prohibits the generation of synthetic big data sets accounting for chemical space with atomistic detail. Exploiting implicit correlations among relaxed structures in training data sets, our machine learning model Graph-To-Structure (G2S) generalizes across compound space in order to infer interatomic distances for out-of-sample compounds, effectively enabling the direct reconstruction of coordinates, and thereby bypassing the conventional energy optimization task. The numerical evidence collected includes 3D coordinate predictions for organic molecules, transition states, and crystalline solids. G2S improves systematically with training set size, reaching mean absolute interatomic distance prediction errors of less than 0.2 Å for less than eight thousand training structures — on par or better than conventional structure generators. Applicability tests of G2S include successful predictions for systems which typically require manual intervention, improved initial guesses for subsequent conventional ab initio based relaxation, and input generation for subsequent use of structure based quantum machine learning models.

[1] Faculty of Physics, University of Vienna, Vienna, Austria. [2] Institute of Physical Chemistry and National Center for Computational Design and Discovery of Novel Materials (MARVEL), Department of Chemistry, University of Basel, Basel, Switzerland. ✉email: anatole.vonlilienfeld@univie.ac.at

The prediction of three-dimensional (3D) structures from a molecular graph is a universal challenge relevant to many branches of the natural sciences. Elemental information and 3D coordinates of all atoms define a system's electronic Hamiltonian, and thereby all related observables which can be estimated as expectation values of approximate solutions to the electronic Schrödinger equation. Energy and force estimates are frequently used to relax the atomic positions on the potential energy surface in order to locate structural minima[1,2]. The many degrees of freedom and various levels of theory for describing potential energy surfaces make structure predictions challenging. The problem is aggravated by the combinatorially large number of possible conformational isomers (cf. Levinthal's paradox[3]), i.e. local minima mapping to the same graph. Often, only low energy conformations are desired, e.g., as practically relevant starting configurations to a chemical reaction[4], or as binding poses in computational drug design[5], requiring conformational scans to identify or rank the most promising representative candidate geometries. While feasible for few and small systems, conformational scans of larger subsets of chemical compound space remain computationally prohibitive.

State of the art approaches for generating 3D molecular structures e.g., ETKDG[6] and Gen3D[7] are very efficient yet carry significant bias since they are based on mathematically rigid functional forms, empirical parameters, knowledge-based heuristic rules, and do not directly improve upon the increase of training data set sizes. While applicable to known and well-behaved regions of chemical compound space, these methods lack generality and are inherently limited when it comes to more challenging systems, such as carbene molecules or transition states (TS). Recent generative machine learning developments might hold promise since they can produce structural candidates to solve inverse molecular design problems[8–12]. Unfortunately, however, they have not yet been used to tackle the 3D structure prediction problem, to the best of our knowledge.

To address the 3D structure with modern supervised learning, we introduce the Graph To Structure (G2S) model. While any other regressor, such as deep neural networks and alike might work just as well, we rely for simplicity on kernel ridge regression (KRR) for G2S in order to predict all elements in the pairwise distance matrix of a single atomic configuration of an out-of-sample molecule or solid. From the pairwise distance matrix, atomic coordinates can easily be recreated. As query input, G2S requires only bond-network and stoichiometry-based information (see Fig. 1a). By exploiting correlations among data-sets free of conformational isomers (restriction to constitutional and compositional isomers only is necessary to avoid ambiguity), G2S learns the direct mapping from chemical graph to that structural minimum that had been recorded in the training data set (which is assumed to be generated in consistent ways), thereby bypassing the computationally demanding process of energy-based conformational search and relaxation.

We have evaluated G2S on QM structures of thousands of constitutional isomers, singlet state carbenes, E2/$S_N$2 transition states (TS), and elpasolite crystals. After training on sufficiently many examples, we find that G2S generated structures for out-of-sample graphs not only have a lower root-mean-square deviation (RMSD) than structures from ETKDG[6] and Gen3D[7] (for the closed-shell molecules for which the latter are applicable) but also exhibit high geometric similarity to the reference quantum chemical structure. Further numerical evidence suggests the applicability of G2S to the prediction problem of transition state geometries, singlet carbene structures, and crystalline solids. We also use G2S to generate coordinates for previously uncharacterized molecules in the QM9 dataset[16] that can be used as input for subsequent QM-based relaxations, or for QML based property predictions. Not surprisingly, analysis of G2S results indicates that interatomic distances between atoms that share strong covalent bonds are easier to learn than between distant atoms which affect each other only through intramolecular non-covalent interactions.

## Results

**G2S performance.** We report G2S performance curves for heavy atom coordinates (not hydrogens) of constitutional isomers, carbenes, TS, and elpasolite structure predictions in Fig. 2. For all data sets and representations studied, root-mean-square deviations of reconstructed geometries of out-of-sample input graphs decrease systematically with training set size. For all QM9 based sets (isomers and carbenes), the bond length and bond hop representations yield systematic improvements with the lowest offset. While bond order exhibits a similar slope, its offset however is markedly higher. This difference is likely due to bond order encoding substantially less explicit information. Note that graph CM and Bag of Bonds (BoB) representation, both yielding better learning curves for atomization energies due to their inverse distance format[17], perform both worse than the bond length representation. Since geometry is directly proportional to distance (and not inversely such as energy), this trend is therefore consistent with the literature findings. The performance of graph CM and BoB for the TS is rather disappointing, but it is in line with trends among machine learning models of the activation energy, already discussed in ref. [18]. If one had to select just one representation, the authors would recommend the bond length representation, which encodes changes in stoichiometry through element-pair specific bond lengths, and which performs best on average (see Table 1 and Fig. 2).

For the TS based performance curves, similar trends are observed with the exception of the bond order representation now resulting in the most accurate G2S model. This is in line with the findings in ref. [18] where the simple one-hot-encoding representation outperforms more physics based representations when it comes to the prediction of activation energies. It is an open question if and how the physics of TS can be properly accounted for within a representation.

From the curves on display in Fig. 2a, it is clear that G2S delivers similar performance no matter if Lewis structures of target systems are well defined or not. For comparison, empirical structure prediction methods ETKDG[7] and Gen3D[6] and have also been applied to the isomer sets (application to carbenes and TS is not possible since these methods are restricted to systems with valid Lewis structure formulas). Their RMSD from QM9 geometries is reached by G2S after training on over 4000 structures. In addition to their quantitative limitations, ETKDG and Gen3D were respectively in 15 and 1.5% of the cases for $C_7O_2H_{10}$, and 6.3 and 19% for $C_7NOH_{11}$ not able to generate a structure from given SMILES at all. This indicates that structure generation can be a challenge for empirical methods, even when it comes to simple closed shells and small organic molecules. Note how the constant slope of the G2S performance curves suggests that even lower prediction errors should be possible for larger training sets. The kinks in the performance curves of the carbene data set result from noise in the DGSOL prediction when solving the distance geometry problem: Actual learning curves of interatomic distances are smooth for all data-sets (see Supplementary Figs. 2–6).

In complete analogy to predicting pairwise atomic distances in molecules, G2S can be trained to predict the pairwise distance of atomic sites in a crystal. Due to the dependence on the size of the unit cell, pairwise distances are predicted in fractional coordinate space instead of Cartesian coordinates, and an additional G2S model is trained to predict the lattice constant, based on the exact

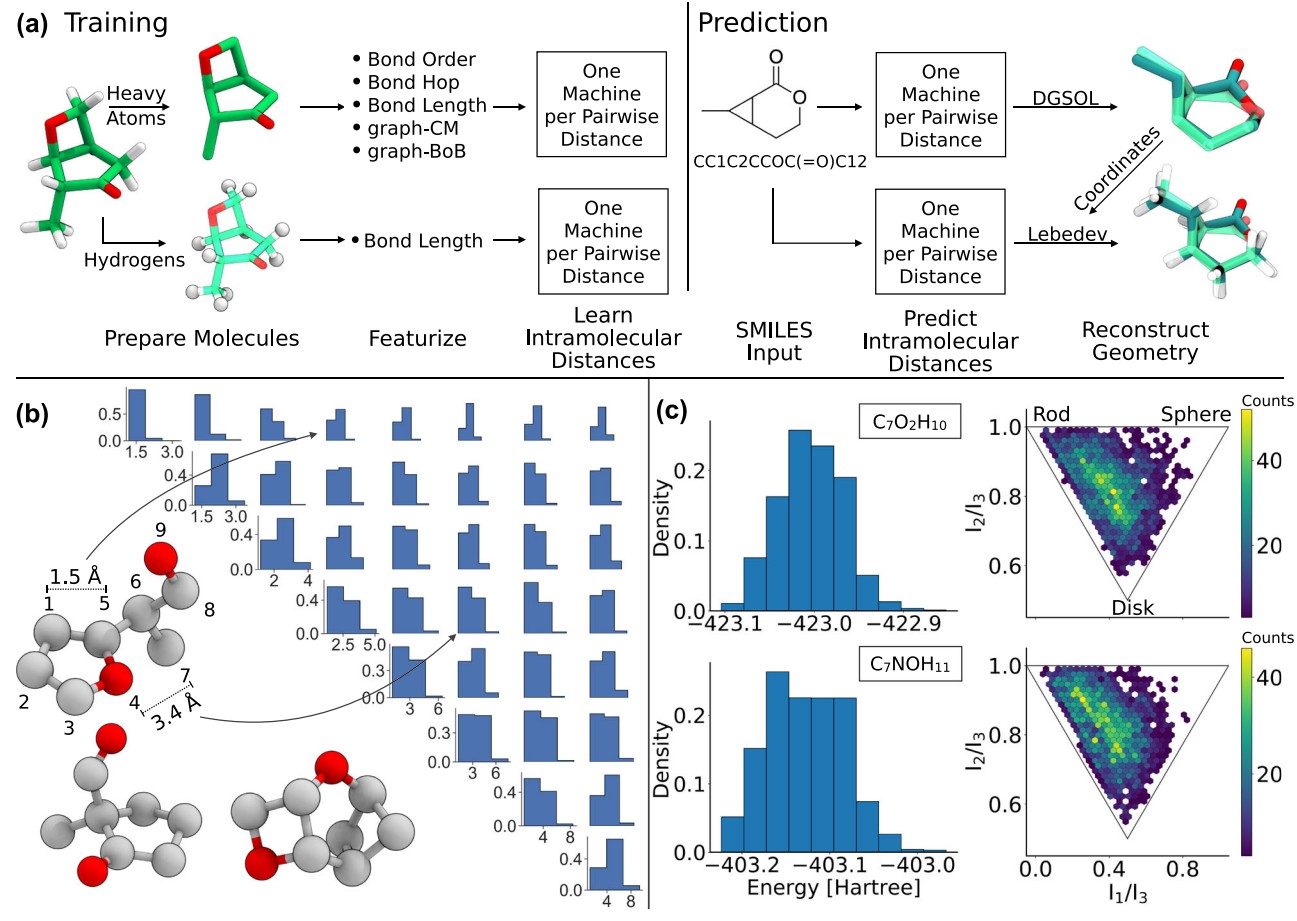

**Fig. 1 Schematic of the G2S workflow and QM9 constitutional isomer dataset. a** From left to right: molecules in the training set are separated into heavy atoms and hydrogens and featurized with a given representation. During training, one machine is used for each pairwise distance. For the prediction of new structures, only molecular connectivity is needed, which can be provided e.g. via SMILES[13] or SELFIES[14]. The machines predict all pairwise distances. The full 3D geometry is then reconstructed using DGSOL[15] for heavy atoms and a Lebedev sphere optimization scheme for hydrogen atoms. **b** Example isomers and distance matrix distributions of the $C_7O_2H_{10}$ QM9 constitutional isomer dataset. The sorting of the atoms and the distance matrix is dependent on the sorting of the molecular representation (example shown for the bond length representation). **c** Energy distribution and principal moments of inertia of the $C_7O_2H_{10}$ and $C_7NOH_{11}$ dataset.

**Table 1 Accuracy of the best performing representation for each dataset at maximum training set size $N^{train}$ and for test set size $N^{test}$ specified.**

|  | $N^{train}$ | $N^{test}$ | MAE [Å] | RMSD [Å] | Representation |
|---|---|---|---|---|---|
| $C_7O_2H_{10}$ | 4876 | 1219 | 0.14 | 0.44 | Bond hop |
| $C_7NOH_{11}$ | 4687 | 1172 | 0.12 | 0.42 | Bond length |
| E2 TS | 1344 | 335 | 0.15 | 0.42 | Bond length |
| SN2 TS | 2228 | 556 | 0.19 | 0.44 | Bond order |
| Carbenes | 4004 | 1002 | 0.13 | 0.38 | Bond length |
| Elpasolite | 8472 | 1528 | 0.16 | 0.15 | FLLA |

Mean absolute error (MAE) of interatomic distances and root-mean-square-deviation (RMSD) calculated for heavy atoms only.

same representation of stoichiometry (FLLA). The performance in Fig. 2b indicates, just as for the molecular cases, systematically decaying prediction errors with growing training set size.

To gain an overview, we also report the best mean absolute and root-mean-square errors for G2S models after training on the largest training set sizes available in Table 1). Mean absolute

errors of less than 0.2Å are obtained in all cases. Exemplary predicted structures, drawn at random and superimposed with their reference, are on display for all molecular data sets in Fig. 3. Visual inspection confirms qualitative to quantitative agreement, the largest deviations corresponding to conformational isomers which can be expected to exhibit small energy differences.

Based on the promising performance of G2S, we have also assessed its performance for 3054 uncharacterized molecules which had failed the QM9 generation protocol[16]. To revisit the problem of predicting the geometries for these uncharacterized molecules, G2S has been trained on 5000 randomly chosen QM9 molecules (varying constitution and composition), and used to predict coordinates for each of them. Subsequent geometry optimization at a B3LYP/6-31G(2df,p) level of theory showed successful convergence of 90% of them (a random selection of unconverged molecules can be seen in Supplementary Fig. 7). A similar success rate has been reached with Gen3D[6] and Open-Babel. Figure 3f depicts randomly drawn structures together with the respective structural formula. At a B3LYP level of theory, 92% of the uncharacterized molecules are expected to converge to a local minimum, which makes G2S a viable initial guess for ab initio structure relaxation[19].

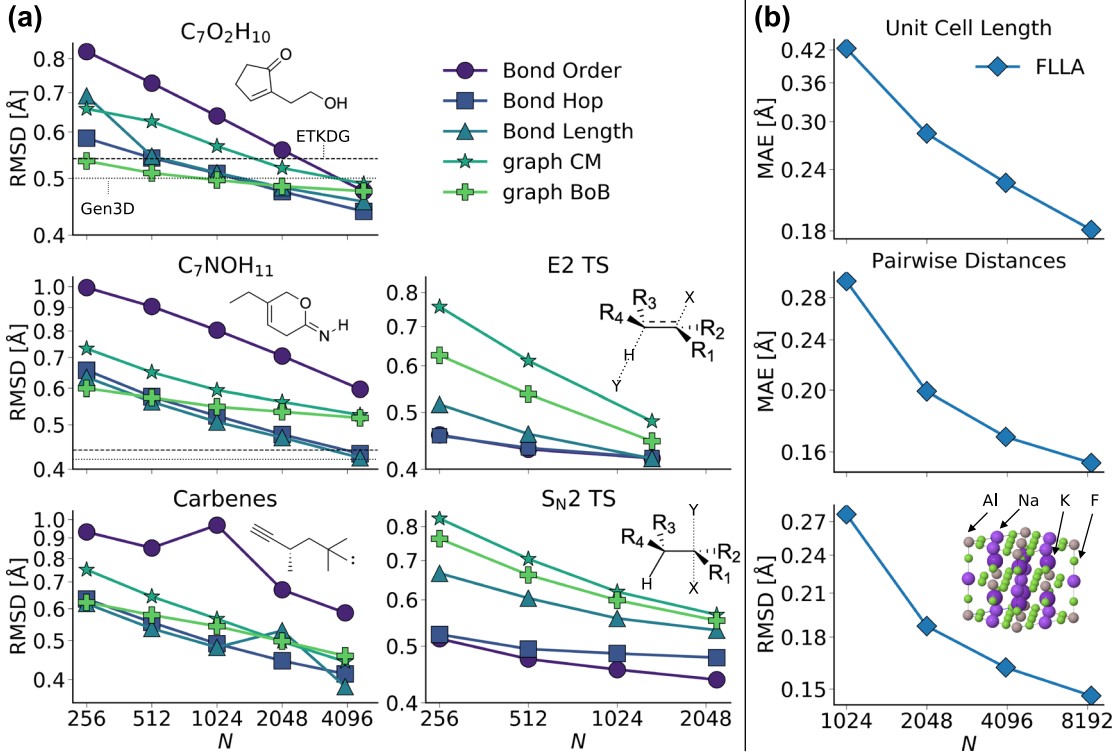

**Fig. 2 Systematic improvement of predictive G2S accuracy with increasing training data for all data sets studied.** Performance curves show mean heavy atom root-mean-square deviation (RMSD) of G2S generated structures (with different representations). **a** Performance curves of isomers, carbenes, and transition states (TS). Insets depict exemplary Lewis structures of each dataset. Horizontal lines show mean RMSD of generated structures with ETKDG[7] and Gen3D[6] from SMILES. **b** Performance curves of the elpasolite dataset using the FLLA representation. Top, mid, and bottom panels depict prediction errors for unit cell length, interatomic distances, and coordinates. The inset illustrates the $AlNaK_2F_6$ elpasolite crystal.

**From G2S output to QM relaxation**. Assessing the usefulness of structure prediction models can be challenging. While from a machine learning perspective, naturally the error is calculated w.r.t. the test dataset (Fig. 4 error type A), energy-based optimization methods are typically evaluated by their deviation from the closest minimum of a higher level of theory structure (Fig. 4 error type B). Since G2S is trained on quantum-based structures, it should inherently be able to predict structures close to the minimum of the used reference method, and should therefore be a useful tool for the automatized generation of meaningful initial structure guesses which can subsequently be used as input in energy-based convergence of the geometry.

We have relaxed the test sets of the $C_7NOH_{11}$ constitutional isomer set and the as $E2/S_N2$ reactant set using G2S output (with bond length representation) as an input for subsequent semiempirical GFN2-xTB[20] for both, as well as DFT (B3LYP) and post-Hartree–Fock (MP2) based relaxation, respectively. The resulting performance curves are shown in Fig. 5 and, again, indicate systematic improvement with training set size, reaching even PM6[21] (semiempirical quantum chemistry) level of theory for error type B of the reactants.

The results for error type A in Fig. 5 (blue curves), however, show that subsequent QM based structure relaxation does not necessarily lead to further improvement for the reactants. While the constitutional isomers improve by almost 0.1 Å, the $E2/S_N2$ reactants tend to get worse. A possible explanation for this counterintuitive trend is that the conformational space of the $E2/S_N2$ reactants is limited to a single dihedral, and once a structure

is predicted by G2S to be in the wrong conformational minimum, further structure relaxation may even increase the error.

Overall, however, G2S predicted input structures result in geometries closer to the minimum of the respective reference method than that of a semiempirical method (Fig. 5 orange curves). In the case of $C_7NOH_{11}$ isomers, the respective error between GFN2-xTB and B3LYP is only 0.13 Å, which could explain an almost equal average distance to both minima. A detailed overview of baseline errors of different methods is given in Supplementary Table 1.

**From G2S output to QML predictions**. The availability of molecular structures is not only a problem for molecular simulations, but also for structure-based machine learning of molecular quantum properties[22]. In order to push the boundary in the exploration of chemical space, either a graph-based model is required, or 3D structures have to be generated. In the case of the latter, the generated structure should be close to the level of theory of the training data in order to avoid large prediction errors. G2S enables us to circumvent this problem by allowing structure-based machine learning models to be trained on predicted structures. Thereby, the property predicting machines learn to compensate the noise of G2S structures, which allows for the future query structures to originate from G2S.

In order to quantify the usefulness of G2S for this problem, we have used G2S output coordinates without further geometry optimization as an input to standard QML representations such as FCHL18[23], FCHL19[24], or BoB[25]. We have focussed on the prediction of atomization and formation energies of constitutional

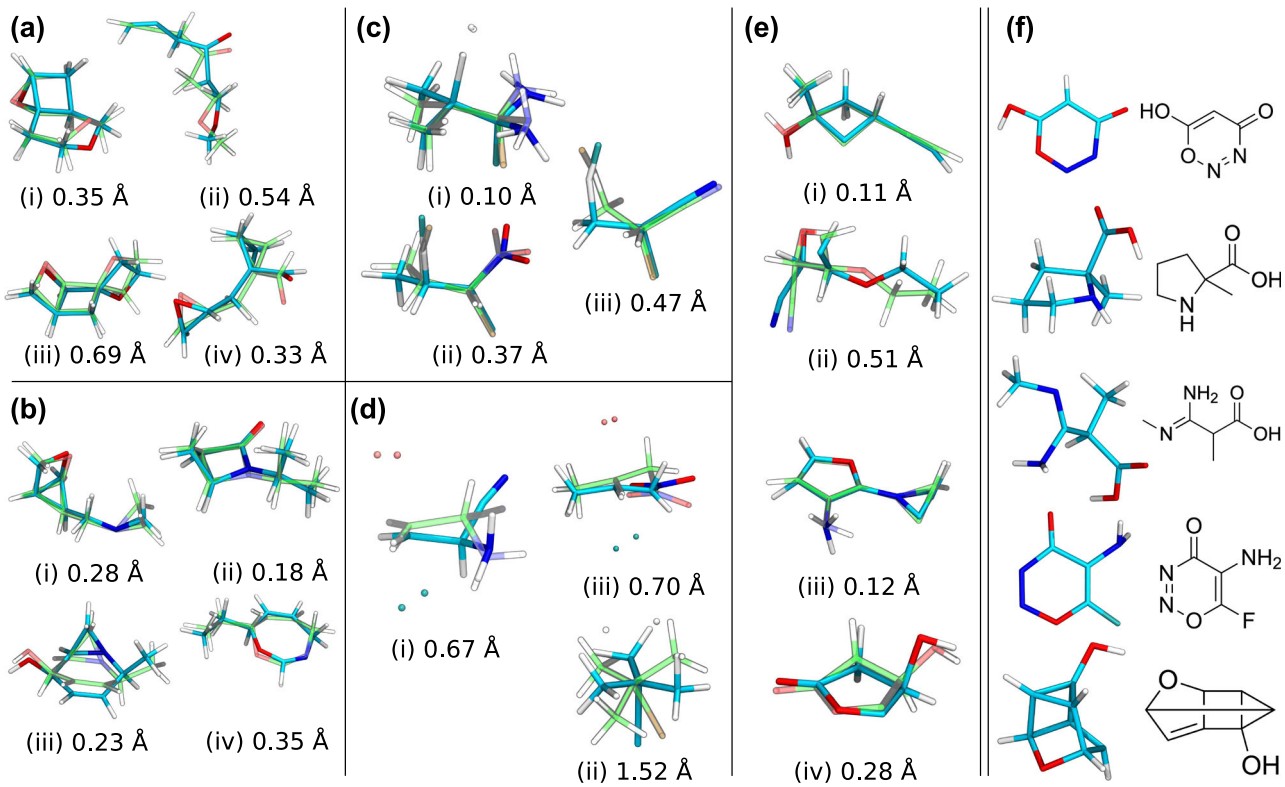

**Fig. 3 Exemplary structures generated with G2S (cyan) for all molecular datasets.** Reference structures are shown in green with corresponding heavy atom root-mean-squared deviation. Panels **a**, **b** constitutional isomers $C_7O_2H_{10}$ and $C_7NOH_{11}$, respectively. **c**, **d** correspond to E2 and $S_N2$ transition states TS with attacking/leaving groups shown as beads, respectively. Panel **e** Carbenes. **f** Five exemplary structures out of the 90% successful predictions of the 3054 uncharacterized QM9 molecules.

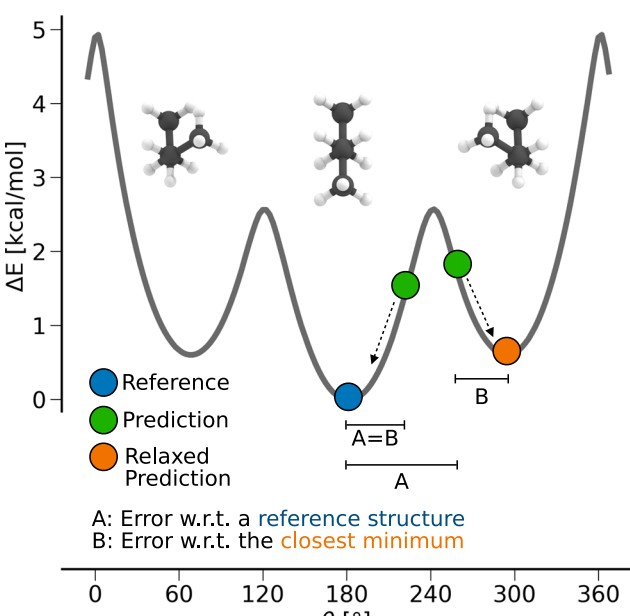

**Fig. 4 Illustration of the structure/conformer prediction problem.** The quality of predicted structures can be quantified in two ways. Error A: the overall accuracy of the machine learning model to reproduce a specific configuration is measured. Error B: Relaxing a predicted structure, the error w.r.t. the closest minimum is calculated, allowing one-to-one comparisons with energy-based structure optimization methods.

isomers and elpasolites, respectively. In Fig. 6, we compare the resulting performance curves to standard QML machines that had access to the "true" reference coordinates as input, as well as to QML machines that used topology only (input graphs for G2S) as input (see Supplementary Fig. 8 for QML learning curves). Again, we note that all performance curves improve systematically with training set size. For atomization energy prediction of $C_7O_2H_{10}$ and $C_7NOH_{11}$ isomers, G2S and FCHL19 still reaches an accuracy of 5 kcal/mol mean absolute error (MAE) at 1024 training points, slowly approaching the coveted chemical accuracy of 1 kcal/mol, and almost matching the accuracy of a DFT structure-based BoB model. Using ETKDG/UFF based geometries as test structures, the performance curves indicate an increasing discrepancy between ETKDG/UFF geometries and energy. The sensitivity of the FCHL19 representation leads, in that regard, to large prediction errors, whereas for small training sizes the BoB representation seems to be more robust. On average, and as one would expect, performance curves improve as one goes from topology only to G2S to QM coordinates as input for QML. The advantage is most substantial for the small training set, in the limit of larger data sets, the performance curves of predictions based on G2S input level off, presumably due to the noise levels introduced by aforementioned error type B, i.e., inherent noise and conformational effects of the predicted structures.

Comparing the impact of the molecular QML representations, the familiar trend that FCHL is more accurate than BoB, is reproduced for either input[22].

For crystals, conformational effects do not exist, which is the reason why FLLA performs almost comparably well to FCHL18 trained on the original structures. Nevertheless, G2S with FCHL18 still reaches an accuracy of 0.3 eV/atom MAE in training set sizes of less than 1000 points.

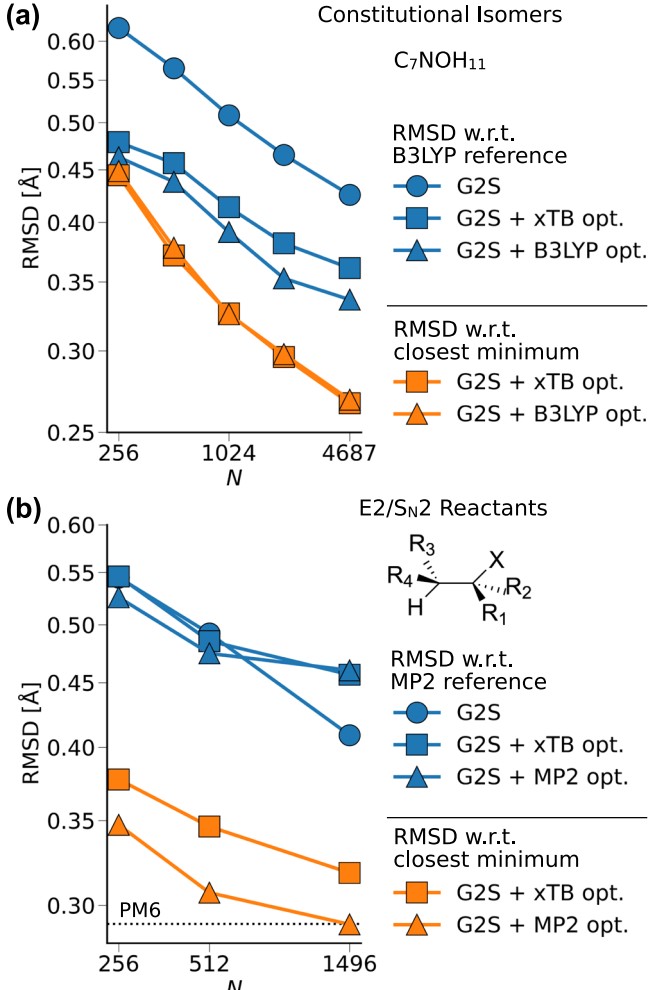

**Fig. 5 Performance curves of G2S predicted structures after subsequent geometry optimization runs.** Blue lines measure the RMSD w.r.t. a quantum-based reference structure (Fig. 4 error type A). Orange lines measure the RMSD w.r.t. G2S predicted structures after a structural relaxation (Fig. 4 error type B). All G2S predictions have been performed using the bond length representation. **a** $C_7NOH_{11}$ constitutional isomers optimized with GFN2-xTB and B3LYP/6-31G(2df,p), respectively. **b** E2/$S_N$2 reactants optimized with GFN2-xTB and MP2/6-311G(d), respectively. The level of theory has been chosen according to the method used in each dataset.

While the G2S based predictions for the training sets specified are not yet comparable to state of the art QML models, an advantage over standard approaches is the generation of new query structures. While 3D structures are available only for a tiny fraction of chemical space, molecular graphs are abundant and can be enumerated systematically[26]. Especially when manual intervention and expensive optimization methods are required, the generation of new target structures becomes almost as difficult as generating the training data itself. In short, a regular QML query requires a structure to be generated with a force field method followed by geometric optimization. Compute times for the respective steps in this workflow are as follows: ETKDG (4 ms), Gen3D (143 ms), GFN2-xTB (257 ms), PM3 (280 ms), DFT (minutes) median timings for $C_7O_2H_{10}$ molecules on a AMD EPYC 7402P CPU). G2S circumvents this procedure by producing structures within 50 ms that can directly be used with a QML model, resulting in orders of magnitude speedups compared to the conventional way (Fig. 6d).

**Analysis and limitations**. The analysis of machine learning predictions is crucial in order to better understand the G2S model. Figure 7 reports the distribution of predicted (largest training set) and reference distances for the $C_7NOH_{11}$ data. We note that, as expected from the integrated results discussed above, the predicted distance distribution overlaps substantially with the respective reference distribution. Small deviations indicate that G2S slightly overestimates covalent bond lengths, and that it underestimates distances to third neighbors. The density differences for second neighbors can hardly be discerned.

The scatter error heat-map plot of predicted versus reference distances (Fig. 7b) indicates the absence of major systematic errors (in line with remarkably good averages), but reveals a larger variance for distances larger than 2.5 Å. A plausible reason for this could be the natural flexibility of molecular structures for flat and long compounds (as opposed to systems dominated by cage-like connectivities). This explanation is corroborated by the trend observed among individual MAE obtained for each distance pair of the distance matrix of $C_7NOH_{11}$ (Fig. 7c): The larger the distance the larger error. As mentioned, the sorting of the representation and distance matrix depends on the norm of the feature values of each row, naturally sorting larger distances to higher indices for the bond length representation. Such prediction errors can then lead to the generation of the wrong conformer, or even diastereomer.

A potential solution could be the decomposition of the full distance matrix into sub-block-matrices containing only close neighbor distances. Conceptually similar to how local atomic representations in QML work, the position of an atom would only depend on the distances of the closest four atoms, allowing its relative position to be uniquely defined. Furthermore, the scalability of G2S would be improved since instead of $n(n-1)/2$ machines for $n$ heavy atoms, only four machines per atom are necessary. Furthermore, since G2S relies on only a single kernel inversion and short representations, the scalability is expected to improve through kernel approximations or efforts in learning efficiency such as the atoms in molecules (AMONS[27]) approach. However, it has to be highlighted that the depicted structures in Fig. 3 have been generated with less than 5000 training molecules available. To that end, the linear trend of the logarithmic learning curves indicates that more data will still improve the accuracy, meaning that fundamentally the learning capability of G2S has not yet achieved its full potential. An improvement in accuracy would make the solution to the distance geometry problem less ambiguous and, therefore, would lead to fewer cases of conformer/diastereomer misclassifications.

In order to further explore the role of the target format, we have also attempted to build machine learning models of entries in the Z-matrix. However, the Z-matrix-based predictions did not improve over the distance matrix-based model estimates (see Supplementary Method I.B.). Possible further strategies to improve on G2S could include Δ-machine learning[28] where deviations from tabulated (or universal force-field based) estimates are modeled.

## Discussion
We have presented G2S, a machine learning model capable of reconstructing 3D atomic coordinates from predicted interatomic distances using bond-network and stoichiometry as input. The applicability of G2S has been demonstrated for predicting structures of a variety of system classes including closed-shell organic molecules, transition state geometries, singlet carbene geometries, and crystal structures. G2S learning curves indicate robust improvements of predictive power as training set size

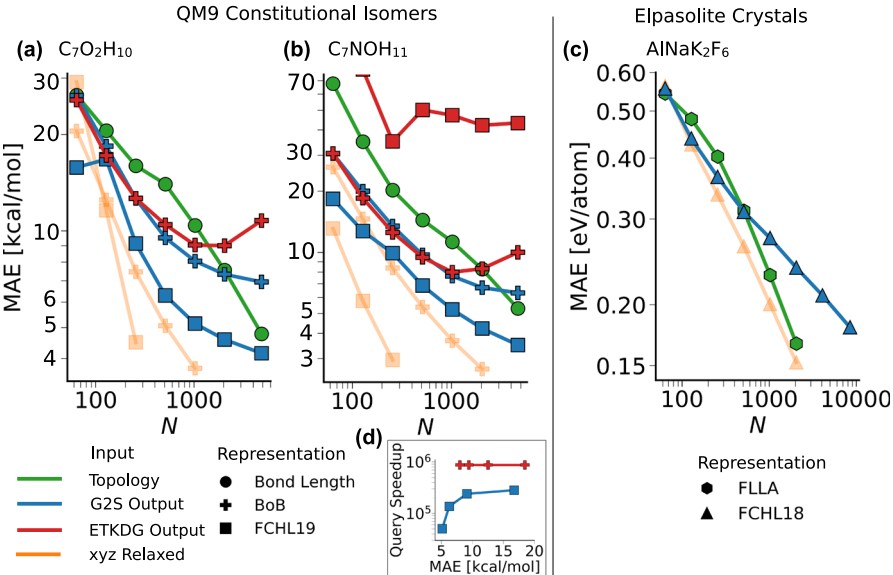

**Fig. 6 Systematic improvement of energy prediction accuracy with increasing training data.** G2S predictions (blue), as well as DFT structures (orange) and ETKDG/UFF structures (red), have been used as inputs to QML models. **a, b** atomization energy prediction of $C_7O_2H_{10}$ and $C_7NOH_{11}$ constitutional isomers, respectively. **c** Prediction of formation energies of elpasolite crystals. **d** Speedup estimate of a G2S (blue) or ETKDG/UFF (red) based QML model over a DFT dependent QML model. This assumes an average of 16 DFT optimization steps required before a structure can be used in QML.

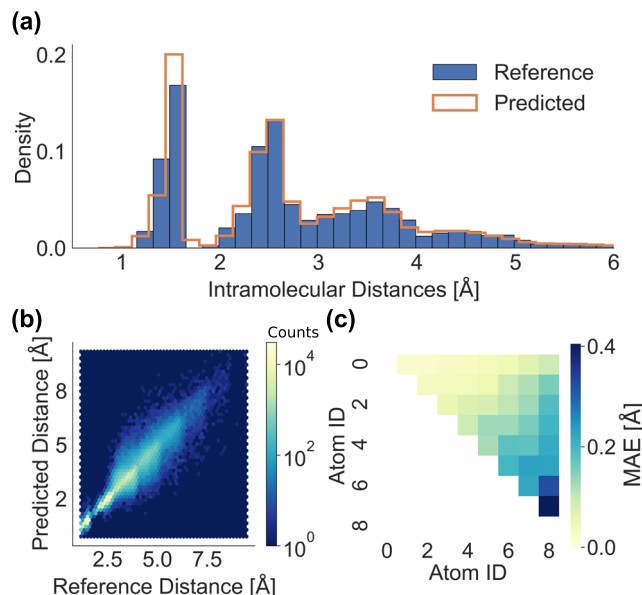

**Fig. 7 Analysis of G2S distance predictions of $C_7NOH_{11}$ constitutional isomers.** For all predictions, G2S was used with the bond length representation and a maximal training set size (4687). **a** Histograms of B3LYP reference (blue) and G2S predicted distances (orange). **b** Hexbin heatmap visualization of B3LYP reference and predicted distances. **c** Heatmap of MAE for each entry of predicted distance matrix.

increases. Training on less than 5000 structures already affords prediction errors of less than 0.2 Å MAE in interatomic distances for out-of-sample compounds—without saturation of the learning curve. We find that G2S predicts chemically valid structures with high geometric similarity towards out-of-sample reference geometries. Our error analysis has identified prediction errors of interatomic distances to be the largest for atoms that are the farthest apart, explaining the possibility of substantial deviations in terms of torsional angles or diastereomers. Comparison to empirical popular structure generators (ETKDG and Gen3D)

indicates that G2S predictions, within their domain of applicability, are on par or better—already for modest training set sizes. We have explored the limits of G2S by also considering geometries of unconventional chemistries such as singlet carbene systems, transition state, or crystalline solids which might be problematic for conventional empirical structure generators. The usefulness of G2S has been illustrated by (a) resolving structures for 90% of the 3054 uncharacterized molecules mentioned in the QM9 database with subsequent ab initio based geometry relaxation, and (b) generating coordinate input for subsequent training of structure-based machine learning predictions of quantum properties, such as atomization energies, reaching prediction errors with hybrid DFT quality.

We believe that a solely data-driven approach is appealing, due to its inherent capability to further improve and generalize across chemical compound space as more training data is being made available. Our extensive numerical evidence suggests that the G2S approach is capable of successfully predicting useful structures throughout chemical compound space and independent of predefined rules or energy considerations. Effectively, G2S accomplishes the reconstruction of atomistic detail from a coarsened representation: The graph of a compound. Our results for elpasolites, transition states, and carbenes already demonstrate that G2S can be trained and applied across differing stoichiometries and sizes. However, given the size and complexity of chemical space, a one fits all solution will just result in a substantially larger model. In that sense, we believe that it is also of significant advantage that G2S adapts already to certain chemical subspaces of interest, and can then be put to good use in that domain. Future work could deal with applications to coarse-grained simulations, Boltzmann averaging or extend above efforts to predict more transition state geometries.

## Methods

**Kernel ridge regression (KRR).** We rely on kernel-based methods which have shown promise in predicting quantum properties throughout chemical compound space after training on sufficient data[22,29–31]. Developed in the 1950s, kernel methods learn a mapping function from a representation vector $x$ to a target property $y$[32,33].

G2S attempts to predict interatomic distances in a sorted distance matrix. The focus on the prediction of internal degrees of freedom facilitates the learning process because of rotational, translational, and index invariance. Note that the subsequent reconstruction of the Cartesian coordinates from a complete set of noisy interatomic distances is straightforward (see below). Within G2S, the interatomic distance target label $y$ between any pair of atoms $I$ and $J$ is defined as

$$y_{IJ}^{G2S}(\mathbf{x}) = \sum_i^N \alpha_i^{(IJ)} \, k(\mathbf{x_i}, \mathbf{x}) \tag{1}$$

with $\boldsymbol{\alpha}_i$ being the $i$-th regression coefficient, $\mathbf{x}_i$ being the representation of the $i$-th molecule in the training set, and $k$ being a kernel function to quantify the similarity of two molecules. The regression coefficients $\alpha$ are obtained from reference interatomic distances $y^{ref}$ according to the standard KRR training procedure.

$$\boldsymbol{\alpha}^{IJ} = (\mathbf{K} + \lambda \mathbf{I})^{-1} \mathbf{y}_{IJ}^{ref} \tag{2}$$

with a regularization coefficient $\lambda$ and the identity matrix $\mathbf{I}$. The regularization strength $\lambda$ is dependent on the anticipated noise in the data and has been determined by hyperparameter optimization. Note that while each interatomic distance matrix element $IJ$ is predicted by a separate G2S model (Eq. 1), formally the training for all models requires only one matrix inversion (Eq. 2). In this sense, G2S represents a single kernel/multi-property KRR model[34].

In practice, we have simply relied on repeated Cholesky decomposition as implemented in QMLcode[35].

To relate the molecular representations via a similarity measure, a kernel function $k$ has to be chosen, as for example,

$$k(\mathbf{x}_i, \mathbf{x}_j) = \exp\left(-\frac{||\mathbf{x}_i - \mathbf{x}_j||_1}{\sigma}\right) \tag{3}$$

$$k(\mathbf{x}_i, \mathbf{x}_j) = \exp\left(-\frac{||\mathbf{x}_i - \mathbf{x}_j||_2^2}{2\sigma^2}\right) \tag{4}$$

Equations (3) and (4) represent Laplacian and Gaussian kernel functions, respectively, a standard choice in KRR based QML[36]. While index-dependent representations can benefit from Wasserstein norms[37], we enforce index invariance by sorting (see below), and have therefore only used either L1 (Laplacian) or L2 (Gaussian) norm.

We optimize the hyperparameters $\sigma, \lambda$ with different choices of kernel function (Gaussian or Laplacian) and representation by using a grid-search and nested fivefold cross-validation. The performance of all models has been tracked in terms of MAE of all distances, as well as RMSD[38–40].

To assess the generalizing capability of G2S for various representations, kernels, and data-sets the test error has been recorded in terms of training set size $N$. The relationship between the test error of a machine learning method in dependence of training set size $N$, a.k.a. learning curve, is known to be linearly decaying on a logarithmic scale[41], which facilitates assessment of learning efficiency and predictive power.

**Graph-based representations**. We use bond order matrices to define molecular graphs, with elements being {0, 1, 2, 3} for bond types none, single, double, and triple, respectively (bond order). For disconnected molecular graphs, e.g., TS, a fully connected graph between attacking/leaving groups and reaction centers is assumed. We have also used a denser way to describe the connectivity of a molecule by counting the number of bonds between atoms following the shortest connecting path (bond hop). These representations capture the connectivity of a molecule but neglect information about atom types. To incorporate atomic information as well as a form of spatial relationship, we weigh the total bond length $l_{ij}$ on the shortest path between atoms i and j by covalent atomic radii taken from refs.[42–44] (bond length). We have introduced more physics (decreasing off-diagonal magnitude with increasing distance) by adapting the Coulomb matrix[29] (CM) representation using the bond length $l$ in the following form,

$$\text{graphCM}_{ij} = \begin{cases} 0.5 Z_i^{2.4}, & i = j, \\ \frac{Z_i Z_j}{l_{ij}}, & i \neq j. \end{cases} \tag{5}$$

with nuclear charges $Z$ (graph CM). The two-body bag form of the CM, BoB[25], was shown to yield improved quantum property machine learning models, and has also been adapted correspondingly for this work (graph BoB). A more detailed description of the representations is provided in Supplementary Methods I.A.

We canonicalize the order of atoms in the representation and distance matrix by sorting the atoms such that $||x_i|| \leq ||x_{i+1}||$ with $x_i$ being the $i$-th row. Due to the use of L1 and L2 norms as metrics in the kernel, the canonicalization process is necessary in order to guarantee that the representation and distance matrix is invariant to the initial order of atoms. Depending on the graph representation, this can lead to an implicit sorting of the distance matrix that is easier to learn, e.g., by sorting short ranges together (Fig. 1b). For the graph BoB representation, distances are ordered similar to the atom-wise binning procedure of BoB.

While the bonding pattern varies for molecules, we presume solids in the same crystal structure to share a fixed adjacency matrix implying that they can solely be described by stoichiometry. The FLLA[45] representation, introduced for Elpasolite

crystals in 2016, exploits this fact by describing each representative site $n$ solely by the row (principal quantum number) and column (number of valence electrons) in the periodic table resulting in an ($2n$-tuple), with sites being ordered according to the Wyckoff sequence of the crystal. In 2017, and using a similar representation, Botti and coworkers have studied the stability of perovskites with great success[46]

**Workflow**. The training of G2S starts with the separation of heavy atoms and hydrogens from the target molecules (Fig. 1a). We generate the heavy atom scaffold first, followed by saturating all valencies with hydrogens. This leads to the scaffold and hydrogen training being independent problems.

After the separation, the input's molecular bonding patterns have to be featurized into a fixed size graph representation. To learn the pairwise distance matrix, we use one model per distance-pair, resulting in $n(n-1)/2$ machines to be trained. This limits the size of any query molecule to at most $n$ heavy atoms (matrices for smaller molecules are padded with zeros). For hydrogens, only the distances to the four closest heavy atom neighbors (not forming a plane) are being considered, requiring four machine learning models. This working hypothesis is consistent with the observation that the deprotonation of small molecules typically only involves local electron density changes[47], making only local geometries predominantly important.

In order to predict interatomic distances for out-of-sample molecules (Fig. 1a), only information about the bonding pattern and nuclear charges is required, e.g., by providing a simplified molecular-input line-entry system[13] (SMILES) or SELFIE[14] string. RDKit is used to generate the corresponding adjacency matrix from which we construct the representation.

To convert the predicted interatomic distances to 3D coordinates, the distance geometry problem[48] has to be solved. For heavy atoms, we use DGSOL[15], a robust distance geometry solver that works with noisy and sparse distance sets.

After reconstructing the heavy atom coordinates, all valencies are saturated by placing hydrogens on a spherical surface provided by a Lebedev[49] sphere. Note that solving the distance geometry problem is independent from G2S, any other approach could have been used just as well.

Regarding the elpasolite crystal structure predictions and in order to allow the conversion from fractional to Cartesian coordinates, an additional machine has been trained to also predict the unit cell length of each stoichiometry. By learning the length of the unit cell with an additional machine, fractional coordinates can then be converted back to Cartesian coordinates.

**Data**. To assess G2S, several quantum-based datasets containing structures of closed shell, singlet carbenes, transition state geometries, as well as elpasolite crystal structures have been considered. The QM9 database[16] has already served as an established benchmark and recently has been used to test generative machine learning models[8–12]. All QM9 molecules were optimized at the B3LYP/6-31G(2df, p)[50–55] level of theory. From QM9, the largest subsets of constitutional isomers, i.e., 6095 and 5859 molecules with $C_7O_2H_{10}$ and $C_7NOH_{11}$ sum formula, respectively, have been selected for this work. Note that already pure constitutional isomers (fixed composition) constitute a difficult target since similar molecular graphs can lead to vastly different 3D geometries. Figure 1 illustrates three exemplary molecules, as well as distance, energy, and moments of inertia distributions for both constitutional isomer sets. As evident from inspection of the latter, the molecular shapes tend to be long and flat with few spherical structures.

In order to push G2S to its limits, systems without well-defined Lewis structures have been considered as represented by two distinct and recent data sets: Carbene and TS geometries. The QMspin[56] database reports over 5'000 singlet and triplet carbene structures (derived through hydrogen abstraction of molecules drawn at random from QM9), for which common structure generation methods would require manual intervention. These structures were optimized using CASSCF[57–59] in a cc-pVDZ-F12[60] orbital basis, and aug-cc-pVTZ[60] density fitting basis. We have used all singlet state carbene structures for the training and testing of G2S.

We have also trained and tested G2S on thousands of TS geometries from the QMrxn20[4] dataset. QMrxn20 consists of $C_2H_6$ based reactant scaffolds, substituted with -NO2, -CN, -CH3, -NH2, -F, -Cl, and -Br functional groups, for which E2/$S_N2$ reaction profiles were obtained using MP2/6-311G(d)[61–65] level of theory.

Regarding solids, we have relied on the elpasolite data-set corresponding to 10,000 training systems made up from main-group elements[45]. All crystal structures had been relaxed using DFT (PBE) with projector augmented wave pseudopotentials[45,66,67].

Finally, we have also extracted the list of 3054 SMILES of "uncharacterized" molecules from the QM9 database, for which the structure generation and B3LYP geometry optimization procedure had led to a mismatch with initial Lewis structures.

**Structure generation and optimization**. The ETKDG[7] method in RDKit version 2019.09.3 and the Gen3D[6] method in Open Babel version 3.0.0 have been used to generate 3D structures from SMILES. As a baseline, B3LYP/6-31G(2df,p) structures of the constitutional isomers and MP2/6-311G(d) E2/$S_N2$ reactants have been optimized with UFF[68], MMFF[69–75], GFN2-xTB[20], and PM6[21], respectively (see Supplementary Table 1). Structure relaxations at the B3LYP/6-31G(2df,p) or MP2/6-311G(d) level of theory have been performed using ORCA version 4.0[76,77]. PM3

and PM6 calculations have been performed using MOPAC2016[78]. If not stated otherwise, no further geometry relaxation with any of the methods has been performed after structures have been generated.

## Data availability

The QM9 constitutional isomer data used in this study is available in the QM9 database at https://doi.org/10.6084/m9.figshare.c.978904.v5. The QMspin and QMrxn databases used in this study are available in the materialscloud database at https://doi.org/10.24435/materialscloud:2020.0051/v1 and https://doi.org/10.24435/materialscloud:sf-tz, respectively. The elpasolite dataset used in this study is available as part of the supplemental information at https://doi.org/10.1103/PhysRevLett.117.135502.

## Code availability

A static implementation of Graph-To-Structure is available at https://doi.org/10.5281/zenodo.4792292. The distance geometry solver DGSOl is available at https://www.mcs.anl.gov/~more/dgsol/.

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

## Acknowledgements

O.A.v.L. acknowledges support from the Swiss National Science Foundation (407540_167186 NFP 75 Big Data). This project has received funding from the European Union's Horizon 2020 research and innovation program under Grant Agreements #952165 and #957189. This project has received funding from the European Research Council (ERC) under the European Union's Horizon 2020 research and innovation program (grant agreement No. 772834). This result only reflects the author's view and the EU is not responsible for any use that may be made of the information it contains. This work was partly supported by the NCCR MARVEL, funded by the Swiss National Science Foundation. Some calculations were performed at sciCORE (http://scicore. unibas.ch/) scientific computing center at the University of Basel.

## Author contributions

D.L. acquired data and wrote new software used in the work. D.L., G.F.v.R., and O.A.v.L. conceived and planned the project, analyzed and interpreted the results, and wrote the manuscript.

## Competing interests

The authors declare no competing interests.
