## [Peer Review File · Nature Communications]

REVIEWER COMMENTS

Reviewer #1 (Remarks to the Author):

Lemm et al. present a kernel-based machine learning (ML) model that is able to reconstruct molecular structures from predicted interatomic distances. The model utilizes structural bond network and stoichiometry as molecular input for the construction of a graph-based molecular representation. 5 different variants were evaluated that consider bonding (Bond Order), atomic connectivity (Bond Hop), Bond Length, as well as graph extension of the popular Coulomb matrices (graph CM) and Bag of Bonds (graph BoB). The graph to structure (G2S) approach provided geometrical structures of closed-shell organic molecules and transition states of simple organic reactions. Overall, I feel that this is a great idea that is surrounded by promising results, but at the end, the manuscript left a bittersweet aftertaste. In particular:

The title should be changed since it does not reflect the accuracy of the G2S model. G2S predicts molecular structures, not "ab initio" structures. The reported 0.10-0.20 Å accuracy for small organic molecules does not reflect an ab initio-quality accuracy. For example, accepted accuracy for DFT-quality bond distances of organic molecules is below 0.02 Å. A more honest title would be "Energy-free machine learning predictions of molecular structures".

Figure 2 includes many interesting points that are not commented in the text. First, why graph CM and BoB are among the most accurate representations for the organic molecules, while they show significantly less accurate results for the E2/SN2 transition states? I was also interested to see a comparison between graph CM/BoBs and their conventional variants. How comparable is the CM/BoB accuracy with respect to graph CM/BoBs? It is also worrisome that different representations (table 1) show increased accuracy for the 5 different examples of Figures 2 and 3. At the end, which is the "best" representation that the user should choose?

As a next step, the authors evaluate the accuracy of G2S by randomly selecting 5000 structures from the QM9 database. The new model provides predicted molecular geometries that are further optimized by DFT with the B3LYP functional, and they report a 90% successful convergence. How was this result validated, what was used as a reference for this claim? Utilization of the G2S-quality structures as input to QM or semi-empirical QM methods provided interesting results. B3LYP has a RMSD of 0.30 Å for organic molecules instead of the expected <0.02 Å (see for example Riley et al. J. Chem. Theory Comput. 2007, 3, 407), while MP2 has a RMSD of 0.45 Å for the E2/SN2 transition states, which is higher than the plain G2S accuracy. Again, what was used as reference for this comparison? Is this source of errors for QM methods due to the decomposed QM geometries? Why only MP2 was used for the E2/SN2 cases and not B3LYP as well? What representation was used for G2S for the data presented on Figure 5 (Bond Order, Bond Hop, Bond Length, graph CM or graph BoB)?

In the analysis and discussion sections, the authors provide suggestions for further development or future targets which might lead to the generation of erroneous structures since basic chemistry principles are omitted. First, they recommend the decomposition of the full distance matrix, which will lead to the learning of the local atomic representation. This approach will eliminate second coordination effects and will either generate artifacts or it will need a larger amount of data for training. Second, they suggest the application of the G2S method for transition metal chemistry. This idea seems scary, since a small geometrical deviation of ~0.1 Å between metal-ligand distances can lead to a different ground spin state with totally different properties. I would rather recommend to delete this statement.

There is no discussion about the QM software(s) that was used in this study (section IV.D). CASSCF is also mentioned in this section, but it was never discussed in the results. For which systems CASSCF was applied, which active space was considered? In addition, CASSCF is far from accurate for molecular structures since dynamical correlation is absent.

Finally, there is no discussion on transferability. I believe that separate models were trained for each molecular case (C7O6H10, C7NOH11, carbenes, E2 and SN2). How G2S can be efficiently used for achieving transferability?

Reviewer #2 (Remarks to the Author):

The manuscript at hand reports a machine learning predictive model (G2S) trained to predict the interatomic distances of chemical systems. The model outperforms other methods in that none is data-derived. The model's accuracy can get as low as 0.2 Å, which sounds promising. As the authors mention, it seems that the main benefit of this approach is to be used as a warm-up geometry optimization for the final QM optimization. The main application of the work could be in the QML predictions, in a way that G2S provides the input structures required for 3D feature representation. However, as it's been illustrated in part C of the results section (Fig. 6), the performance is about 20% worse than the predictions with DFT structures. Thus, I request the authors to add one learning curve for the QML training with structures from the UFF method. I understand UFF can be somewhat more expensive than G2S, but let's have its learning curve as the baseline of any graph2structure-like predictive models that will most probably emerge soon. In addition, the authors discuss how the model can benefit from more training data from the chemical compound space, but I am not quite sure if a kernel method that is not easily scalable can be of great help in the scale of molecular space.

Overall, the proposed approach is an interesting first step towards structure prediction. Still, I am not convinced of the significance of the performance and application of the work based on the current results. The addition of the QML predictions using UFF optimized structures and corresponding cost-benefit analysis may help derive a robust conclusion.

Reviewer #3 (Remarks to the Author):

This manuscript presents a novel approach to computing 3D structures of molecules from connectivity information, such as that present in a SMILES string. The approach avoids searching for the minimum of an energy function by instead: (1) using an ML model to predict inter-atomic distances from molecular graphs and (2) using existing methods (e.g. DGSOL) to compute the 3D structure from these estimated distances.

The approach is tested on well-chosen datasets and the results are promising. For conformers of organic molecules, the approach leads to geometries that are better (as measured by RMSD from quantum chemical structures) than standard methods (ETKDG And Gen3D). The results for the transition states of SN2 and E2 reactions are especially promising, because these structures are not amenable to standard methods that rely on a simple Lewis structure. The relatively small amount of data needed to train the models is also an advantage, especially for transition states where it is difficult to obtain large sets of training data.

The manuscript presents a novel approach to a challenging and important problem, that of quickly generating approximate molecular structures. The results are promising and the manuscript is well written and concise. (Figs. 4 and 5, in particular, are a nice way to summarize these results.) I believe the results would be of interest to a broad audience.

I do have a few questions, and suggestions for improving the presentation.

The approach develops a separate model for each entry in a sorted list of interatomic distances in the molecule. When training on molecules with up to n heavy atoms, this leads to $n*(n-1)/2$ models. Does this mean the resulting G2S can not make predictions for molecules with more than n heavy atoms? (Alternatively, it may be that DGSOL can make reasonable predictions of 3-D structure, given only the estimates for the $n(n-1)$ closest distances in the molecule.)

The manuscript seems to imply that the computational cost of G2S is low, compared to searching for minima on an energy surface. For the size of molecules explored here, can the authors provide an estimate of this cost (e.g. in comparison to ETKDG or Gen3D, or minimizations in a low-cost quantum method such as PM3)?

The features input to the model must be derived from only a graph of the molecular structure. It would be helpful to have, possibly in the SI, additional details on how the features listed in the manuscript were constructed (e.g. the definition of the graph CM and graph BOB features).

We would like to thank all reviewers and the editor for the constructive criticism and for contributing to the overall improvement of the quality of the manuscript. In the following we will provide a point-by-point response to each of the reviewers' comments and refer to the corresponding changes to the revised MS.

Reviewer: 1

Reviewer Comment:

Overall, I feel that this is a great idea that is surrounded by promising results, but at the end, the manuscript left a bittersweet aftertaste.

Authors' response:

We thank the reviewer for investing time into providing us with valuable feedback for our work. We also appreciate the recognition that the underlying idea of this study is 'great', and supported by the results.

Reviewer Comment:

The title should be changed since it does not reflect the accuracy of the G2S model. G2S predicts molecular structures, not "ab initio" structures. The reported 0.10-0.20 Å accuracy for small organic molecules does not reflect an ab initio-quality accuracy. For example, accepted accuracy for DFT-quality bond distances of organic molecules is below 0.02 Å. A more honest title would be "Energy-free machine learning predictions of molecular structures".

Authors' response:

We thank the reviewer for the suggestion. The reviewer is correct that the old title could have been misunderstood in the sense that a reader would expect a very high-quality structure prediction. What we meant, of course, was to indicate that the machine learning model infers structures from 'ab initio' data, meaning that the reference data used for training and test has resulted from quantum mechanical calculations. In this sense, any 'ab initio' method, for example also a semi-empirical quantum method, could have been used for this. Furthermore, our learning curves indicate that if more data had been available, also more accurate geometries could have been obtained. A third reason for choosing this term is that we wanted to convey the fact that also reactive states (transition geometries) or structures of open-shell systems can be considered.

To account for the reviewer's point and to avoid the false impression that our predicted structures are extremely high-quality, we have changed the title as follows:

Machine learning based energy-free structure predictions of molecules (closed and open-shell), transition states, and solids

Reviewer Comment:

Figure 2 includes many interesting points that are not commented in the text. First, why graph CM and BoB are among the most accurate representations for the organic molecules, while they show significantly less accurate results for the E2/SN2 transition states?

Authors' response:

We thank the reviewer for highlighting this point. In the submitted manuscript the following is already stated regarding this point

Note that graph CM and BoB representation, both yielding better learning curves for atomization energies due to their inverse distance format, perform both worse than Bond Length. Since geometry is directly proportional to distance (and not inversely such as energy), this trend is therefore consistent with the literature findings

It is worth mentioning that while graph-CM/BoB are performing okayish for $C_7O_2H_{10}$ and carbenes, they are significantly worse for C_7NOH_{11} constitutional isomers. Apart from the inverse scaling being better suited for learning of energies, it has also already been discussed in Ref. [20] in the revised version that even more sophisticated representations well suited for minimum structures yield much worse performance when it comes to transition states. As such, performance trends among representations, while much investigated and explored for minimum energies, seem to be less well understood when it comes to other properties such as interatomic distances of highly distorted systems (such as transition states). In order to expand on this point, we have now also inserted the following sentence in the revised version just below the text just quoted:

The performance of graph CM and BoB for the transition states is rather disappointing, but it is in line with trends among machine learning models of the activation energy, already discussed in Ref.[20]

Reviewer Comment:

How comparable is the CM/BoB accuracy with respect to graph CM/BoBs?

Authors' response:

Ordinary CM/BoB would require knowledge of coordinates as an *input*. Since this study is about predicting coordinates (i.e., coordinates are *output*), only graph input should be used. For this reason, conventional CM/BoB are not applicable as representations.

Reviewer Comment:

It is also worrisome that different representations (table 1) show increased accuracy for the 5 different examples of Figures 2 and 3. At the end, which is the “best” representation that the user should choose?

Authors' response:

All the different data-sets in Figure 2 differ substantially in their nature. Since each of them is rather small in size (for fully converged machine learning models typically much larger data sets are required) it is not surprising that trends among representations differ for machine learning models trained on different data sets. The most important point, however, is that prediction errors improve with training set size for all models. In practice, a user can always try various representations (due to the small training set sizes generating the learning curve is rather inexpensive), and select the best performing representation for the data-set at hand. If pressed for selecting a single representation, however, we would recommend the 'Bond-length' representation which (a) yields very low prediction errors for the largest training set sizes for

all sets but for the SN2 data set, and which (b) can account for varying stoichiometry due to the implicit element-dependence of bond-lengths. A corresponding statement has now been inserted in the relevant text of the revised manuscript (Sec II A):

If a user had to select just one representation, the authors would recommend the Bond-length representation which encodes changes in stoichiometry through element-pair specific bond-lengths, and which performs best on average (see Tab 1 and Fig. 2).

Reviewer Comment:

The new model provides predicted molecular geometries that are further optimized by DFT with the B3LYP functional, and they report a 90% successful convergence. How was this result validated, what was used as a reference for this claim?

Authors' response:

As already stated at the end of section II A, G2S has been used to predict geometries of unresolved molecules. The result was validated by using this G2S output as an input for subsequent structure relaxation using the B3LYP/6-31g(2df,p) [same level of theory as it had been used in QM9]. After the B3LYP relaxation, the converged structures have been checked and 90% showed successful relaxation convergence as well as no broken bonds (as observed before with this subset). We consider the B3LYP geometry convergence to be sufficient as a validation of the G2S predictions.

Note that, and as also pointed out in that very same paragraph, a recent study troubleshot these unstable structures using different level of theories (force field, B3LYP, ω B97XD, CCSD).[1] Senthil et al. showed that at a B3LYP/6-31g(2df,p) level of theory, only 92% of the unresolved molecules can be considered stable, meaning that G2S was able to stabilize almost all that could be considered stable. Ring openings of these molecules can occur due to a $-NNO-$ moiety, of which we listed some examples in the supplementary. Unfortunately, the data of these troubleshot molecules is not available to us since these authors will be investigating them in the near future.

Reviewer Comment:

Utilization of the G2S-quality structures as input to QM or semi-empirical QM methods provided interesting results. B3LYP has a RMSD of 0.30 Å for organic molecules instead of the expected < 0.02 Å (see for example Riley et al. J. Chem. Theory Comput. 2007, 3, 407), while MP2 has a RMSD of 0.45 Å for the E2/SN2 transition states, which is higher than the plain G2S accuracy. Again, what was used as reference for this comparison? Is this source of errors for QM methods due to the decomposed QM geometries?

Authors' response:

We thank the reviewer for highlighting this point. In Figure 5, two different comparisons are being made. The blue curves show the RMSD of a G2S prediction w.r.t. the original reference structure (out-of-sample test set) G2S was supposed to predict. If G2S predicts a geometry which is in the same attractive basin as the reference structure (Fig. 4 blue ball), structure relaxation will lead to an RMSD of around zero. If G2S predicts a different conformer, the RMSD will be larger than zero. For both examples, the constitutional isomers and E2/SN2 reactants, the RMSD is non-zero which indicates that G2S predicted different conformers.

However, since the learning curves are linearly decreasing with increased training set size, more training data will improve the amount of correctly predicted conformers. The orange curves of Figure 5 show the RMSD of a G2S prediction w.r.t. the closest local minimum which allows to assess the accuracy w.r.t. a higher level of theory method better. In the revised manuscript, we adjusted the colors and description in Figure 4 and Figure 5 to better highlight this difference.

Figure 4: Illustration of the structure prediction problem on a butane dihedral energy profile (GFN2-xTB). The quality of predicted structures can be quantified in two ways. Error A: the overall accuracy of the machine learning model to reproduce a specific configuration is measured. Error B: Relaxing a predicted structure, the error w.r.t. the closest minimum is calculated, allowing one-to-one comparisons with energy based structure optimization methods.

Figure 5: Performance curves of G2S (Bond Length representation) predicted input coordinates of C_7NOH_{11} constitutional isomers and E2/SN2 reactants after subsequent *ab initio* based geometry optimization runs. Blue lines measure the RMSD w.r.t. a quantum-based reference structure (Fig. 4 error type A). Orange lines measure the RMSD w.r.t. G2S predicted structures after a structural relaxation (Fig. 4 error type B). (a) C_7NOH_{11} constitutional isomers optimized with GFN2-xTB and B3LYP/6-31G(2df,p), respectively. (b) E2/SN2 reactants optimized with GFN2-xTB and MP2/6-311G(d), respectively. The level of theory has been chosen according to the method used in each dataset.

Reviewer Comment:

Why only MP2 was used for the E2/SN2 cases and not B3LYP as well?

Authors' response:

The level of theory has been chosen according to the level of theory that has been used in the respective dataset taken from the literature. For this manuscript, we did not produce any new data ourselves. But in order to be consistent, the constitutional isomers from the QM9 dataset have been optimized with B3LYP whereas the E2/SN2 reactants have been optimized with MP2. We have added this clarification to the caption of Figure 5 (see above).

Reviewer Comment:

What representation was used for G2S for the data presented on Figure 5 (Bond Order, Bond Hop, Bond Length, graph CM or graph BoB)?

Authors' response:

For both cases, C₇NOH₁₁ constitutional isomers and E2/SN2 reactants, the bond length representation has been used to predict the structures. We have added this clarification to the caption of Figure 5 (see above).

Reviewer Comment:

First, they recommend the decomposition of the full distance matrix, which will lead to the learning of the local atomic representation. This approach will eliminate second coordination effects and will either generate artifacts or it will need a larger amount of data for training.

Authors' response:

We agree that a local decomposition of the full distance matrix could lead to artifacts. However, the training data needed would not increase since local environments such as rings or functional groups are ubiquitous in chemical space and could be learned even when the data availability is sparse.

Reviewer Comment:

Second, they suggest the application of the G2S method for transition metal chemistry. This idea seems scary, since a small geometrical deviation of 0.1 Å between metal-ligand distances can lead to a different ground spin state with totally different properties. I would rather recommend to delete this statement.

Authors' response:

We thank the reviewer for the suggestion and have removed this statement.

Reviewer Comment:

There is no discussion about the QM software(s) that was used in this study (section IV.D).

Authors' response:

We thank the reviewer for highlighting this and moved the section about structure generation

and optimization that includes software from the supplementary to the method section (IV.E). The subsection reads as follows:

The ETKDG method in RDKit version 2019.09.3 and the Gen3D method in Open Babel version 3.0.0 have been used to generate 3D structures from SMILES. As a baseline, B3LYP/6-31G(2df,p) structures of the constitutional isomers and MP2/6-311G(d) E2/SN2 reactants have been optimized with UFF, MMFF, GFN2-xTB and PM6, respectively (see SI). Structure relaxations at the B3LYP/6-31G(2df,p) or MP2/6-311G(d) level of theory have been performed using ORCA version 4.0. PM3 and PM6 calculations have been performed using MOPAC2016. If not stated otherwise, no further geometry relaxation with any of the methods has been performed after structures have been generated.

Reviewer Comment:

CASSCF is also mentioned in section IV.E, but it was never discussed in the results. For which systems CASSCF was applied, which active space was considered? In addition, CASSCF is far from accurate for molecular structures since dynamical correlation is absent.

Authors' response:

CASSCF was used for singlet state carbenes during the generation of the QMSpin dataset described in Ref.[2]. The active space that has been considered in their study is a two electrons in two orbitals $[(2e,2\sigma)]$ active space. For this study we did not perform additional CASSCF calculations.

Reviewer Comment:

Finally, there is no discussion on transferability. I believe that separate models were trained for each molecular case (C7O2H10, C7NOH11, carbenes, E2 and SN2). How G2S can be efficiently used for achieving transferability?

Authors' response:

The reviewer is correct that independent models have been trained for each dataset. Of course, it is possible to train a single model on all combined data sets. We have included the following statement on transferability in the Discussion section:

Our results for elpasolites, transition states, and carbenes already demonstrate that G2S can be trained and applied across differing stoichiometries and sizes. However, given the size and complexity of chemical space, a one fits all solution will just result in a substantially larger model. In that sense, we believe that it is also of significant advantage of G2S that it adapts already to certain chemical subspaces of interest, and can then be put to good use in that domain.

Reviewer: 2

Reviewer Comment:

The manuscript at hand reports a machine learning predictive model (G2S) trained to predict the interatomic distances of chemical systems. The model outperforms other

methods in that none is data-derived. The model’s accuracy can get as low as 0.2 Å, which sounds promising. As the authors mention, it seems that the main benefit of this approach is to be used as a warm-up geometry optimization for the final QM optimization. The main application of the work could be in the QML predictions, in a way that G2S provides the input structures required for 3D feature representation.

Authors’ response:

We thank the reviewer for investing time into providing us with valuable feedback for our work. We also appreciate the recognition that our approach is promising and useful.

Reviewer Comment:

It’s been illustrated in part C of the results section (Fig. 6), the performance is about 20% worse than the predictions with DFT structures. Thus, I request the authors to add one learning curve for the QML training with structures from the UFF method. I understand UFF can be somewhat more expensive than G2S, but let’s have its learning curve as the baseline of any graph2structure-like predictive models that will most probably emerge soon.

Authors’ response:

We would like to thank the reviewer for this interesting suggestion. Using the same input for ETKDG as for G2S, we have used UFF to relax the coordinates for additional QML learning curves of energies in Fig 6 (see below). For larger training set sizes, the UFF based learning gets worse, indicating an increasing discrepancy between UFF geometry outputs and energy. For C7O2H10, FCHL based predictions using UFF geometries as an input are even so bad that they are off the chart. The following text has been included in the discussion of this figure:

Using ETKDG/UFF based geometries as test structures, the performance curves indicate an increasing discrepancy between ETKDG/UFF geometries and energy. The sensitivity of the FCHL19 representation leads, in that regard, to large prediction errors, whereas for small training sizes the BoB representation seems to be more robust.

Reviewer Comment:

In addition, the authors discuss how the model can benefit from more training data from the chemical compound space, but I am not quite sure if a kernel method that is not easily scalable can be of great help in the scale of molecular space.

Authors’ response:

We thank the author for making this interesting point. It could be expected that given the size of chemical space, any structure generation method will struggle. Especially for empirical and rule based methods such as ETKDG or Gen3D it will be a problem due to the manual work required to extend existing parameters, which is not scalable. G2S is not attempting to be a one fits all solution, but rather be an alternative method that can be tailored towards a user’s need. Regarding the performance of kernel methods with large amounts of data, the G2S method effectively relies on only a single kernel and short representations, making it significantly faster than regular kernel based QML models that use atomic local environments. In the past, it has already been shown that kernel methods can easily be trained on

Figure 6: Systematic improvement of energy prediction accuracy with increasing training data using G2S predictions (blue) as well as DFT structures (orange) and ETKDG/UFF structures (red) as an input to QML models. (a) and (b) atomization energy prediction of C₇O₂H₁₀ and C₇NOH₁₁ constitutional isomers, respectively. (c) Prediction of formation energies of elpasolite crystals. (d) Speedup estimate of a G2S (blue) or ETKDG/UFF (red) based QML model over a DFT dependent QML model. This assumes an average of 16 DFT optimization steps required before a structure can be used in QML.

training set sizes up to 100'000 points. Moreover, developments on kernel approximations have been shown to be applicable to dataset sizes with millions of datapoints [3]. In addition, constant efforts in learning efficiency and dataset reduction through, for example the atoms in molecules (AMONS) approach [4], are expected to reduce the effective dimensionality throughout compound space, and thereby increase kernel based data-efficiency to such a degree that scalability can be achieved for many relevant applications. It should be said, however, that the underlying idea of G2S, to map a graph to interatomic distances, could have also been learned with other, more scalable, regressor functions, such as neural networks or random forests. Only future work will tell which specific variant is the most preferable.

To reflect this discussion within our revised manuscript, we have included the following sentence in section II D.:

Furthermore, since G2S relies on only a single kernel inversion and short representations, the scalability is expected to improve through kernel approximations or efforts in learning efficiency such as the atoms in molecules (AMONS) approach.

Reviewer Comment:

Overall, the proposed approach is an interesting first step towards structure prediction. Still, I am not convinced of the significance of the performance and application of the work based on the current results. The addition of the QML predictions using UFF optimized structures and corresponding cost-benefit analysis may help derive a robust conclusion.

Authors' response:

We thank the reviewer for the comments and suggestions, and we hope that the addition, as requested, of the UFF baseline results now better illustrates the promise of G2S. In terms of significance and performance, we would like to reiterate that already for such modest training set sizes, G2S is more reliable than rdkit, the most popular cheminformatics package in the community. This resonates in the fact that rdkit is unlikely to include many more chemistries due to the substantial manual effort required to develop suitable algorithms, while G2S solely requires additional training data which regularly is generated in an automated fashion. In this sense, G2S requires compute resources, not human effort to yield even better predictions. Furthermore, G2S is more general than equally efficient approaches such as ETKDG or Gen3D which fail to get anything for cases such as carbenes, transition states, or crystals. Conceptionally, we also believe that this is the first work to exploit ML to directly side-step the iterative (and sometimes lengthy) energy (gradient) based optimization procedures, implemented in virtually every atomistic simulation package on this planet. As such, we do believe that our work is highly relevant and of tremendous potential impact.

Reviewer: 3

Reviewer Comment:

The manuscript presents a novel approach to a challenging and important problem, that of quickly generating approximate molecular structures. The results are promising and the manuscript is well written and concise. (Figs. 4 and 5, in particular, are a nice way to summarize these results.) I believe the results would be of interest to a broad audience.

Authors' response:

We thank the reviewer for investing time into providing us with valuable feedback for our work. We greatly appreciate the recognition that our results are promising and of broad interest.

Reviewer Comment:

When training on molecules with up to n heavy atoms, this leads to $n*(n-1)/2$ models. Does this mean the resulting G2S can not make predictions for molecules with more than n heavy atoms? (Alternatively, it may be that DGSOL can make reasonable predictions of 3-D structure, given only the estimates for the $n(n-1)$ closest distances in the molecule.)

Authors' response:

The reviewer is correct that within our current implementation the size of a new query molecule is limited by the largest molecule in the training set. We modified the statement in section IV.C to make this limitation more clear:

To learn the pairwise distance matrix, we use one model per distance-pair, resulting in $n(n - 1)/2$ models to be trained. This limits the size of any query molecule to at most n heavy atoms (matrices for smaller molecules are padded with zeros).

It is definitely possible for DGSOL to work with approximate distances or boundaries and will be investigated in future studies. We believe, however, that this is not a fundamental limitation. Currently, we are working on a localized and scalable improved version of G2S which will no longer suffer from this restriction.

Reviewer Comment:

The manuscript seems to imply that the computational cost of G2S is low, compared to searching for minima on an energy surface. For the size of molecules explored here, can the authors provide an estimate of this cost (e.g. in comparison to ETKDG or Gen3D, or minimizations in a low-cost quantum method such as PM3)?

Authors' response:

The total cost to predict a structure with G2S is in the order of a few milliseconds. The current bottleneck of our G2S implementation is the conversion of distance matrix to 3D coordinates with DGSOL (not our code) which consumes 85% of the time. We added a speed comparison of G2S, ETKDG, Gen3D structure generation and GFN2-xTB, PM3 minimization on the C7O2H10 isomers to the end of section II.C in the updated manuscript.

In short, a regular QML query requires a structure to be generated with a force field method followed by geometric optimization. Compute times for the respective steps in this work-flow are as follows: ETKDG (4 ms), Gen3D (143 ms), GFN2-xTB (257 ms), PM3 (280 ms), DFT (minutes) median timings for C₇O₂H₁₀ molecules on a AMD EPYC 7402P CPU). G2S circumvents this procedure by producing structures within 50 ms that can directly be used with a QML model, resulting in orders of magnitude speedups compared to the conventional way (Fig. 6 (d)).

Reviewer Comment:

The features input to the model must be derived from only a graph of the molecular structure. It would be helpful to have, possibly in the SI, additional details on how the features listed in the manuscript were constructed (e.g. the definition of the graph CM and graph BOB features).

Authors' response:

We thank the reviewer for the suggestion and have added a section on bond length based representations (Bond Length, graph CM, graph BoB) in the SI:

To construct representations from a molecular graph that rely on bond lengths, covalent atomic radii are required for each type of bond (single, double, triple). Using the atomic radii as weights, the bond length distance or shortest path l_{ij} between two atoms i and j in a graph is calculated using Dijkstra's algorithm as implemented in `igraph`. Calculating the bond length distances for all atom pairs in a molecule results in a representation of the following form:

$$\text{Bond Length}_{ij} = \begin{cases} 0, & i = j, \\ l_{ij}, & i \neq j. \end{cases} \quad (1)$$

with l_{ij} being the bond length distance/shortest path between the atoms i and j . To include more physics, the bond length distance can be used to approximate 2-body interactions that are commonly used in QML representations such as the Coulomb Matrix (CM) or Bag-of-Bonds (BoB). The CM representation contains the coulomb interaction scaled by the interatomic distance as off-diagonal elements, while the diagonal represents an approximation to the atomic energy of the nuclear charge Z_i . This leads to a representation with the following form:

$$\text{CM}_{ij} = \begin{cases} 0.5Z_i^{2.4}, & i = j, \\ \frac{Z_i Z_j}{|\mathbf{R}_i - \mathbf{R}_j|}, & i \neq j. \end{cases} \quad (2)$$

with nuclear charge Z and atomic coordinates \mathbf{R} . Since the atomic coordinates are not available for a structure prediction task, the representation has to be adapted for molecular graphs. The bond length distance approach described above suits as an approximation to the intermolecular distance and can therefore be used to adapt the off-diagonal term of the CM to work in a graph setting. The adapted representation, dubbed graph CM, has the following form:

$$\text{graph CM}_{ij} = \begin{cases} 0.5Z_i^{2.4}, & i = j, \\ \frac{Z_i Z_j}{l_{ij}}, & i \neq j. \end{cases} \quad (3)$$

with nuclear charge Z and bond length distance l_{ij} . To convert the CM into a BoB representation, the CM has to be vectorized by grouping all matrix terms into specific bins. The thereby created canonical order (bag of bonds) ensures that during the kernel calculation only similar bins are compared. Each bin describes a particular bond type (H-H, C-C, C-H etc.). In this regard, the BoB and graph BoB representation use the same components as their respective matrix counterpart (CM and graph CM), but only differ through transforming the matrix into a canonical vector. Since the distance matrix is sorted based on the sorting of the representation, the distance matrix undergoes the same vectorization and binning procedure as the graph BoB representation.

References

- [1] S. Senthil, S. Chakraborty, and R. Ramakrishnan, "Troubleshooting unstable molecules in chemical space," *Chemical Science*, vol. 12, no. 15, pp. 5566–5573, 2021.
- [2] M. Schwilk, D. N. Tahchieva, and O. A. von Lilienfeld, "Large yet bounded: Spin gap ranges in carbenes," 2020.
- [3] G. Meanti, L. Carratino, L. Rosasco, and A. Rudi, "Kernel methods through the roof: handling billions of points efficiently," 2020.

- [4] B. Huang and O. A. von Lilienfeld, “Quantum machine learning using atom-in-molecule-based fragments selected on the fly,” *Nature Chemistry*, vol. 12, pp. 945–951, Sept. 2020.

REVIEWERS' COMMENTS

Reviewer #1 (Remarks to the Author):

The authors have adequately addressed my comments and provided substantial edits in the revised manuscript. Together with the additional clarifications that were provided for the other two reviewers, the current manuscript is more transparent and argumentation on the promising new method justifiable. I do not have any further comments.

Reviewer #2 (Remarks to the Author):

All my concerns are addressed in the revised version of the article, and I also believe the results are promising and impactful. I have no additional comments.

We would like to again thank all reviewers and the editor for the constructive criticism and for contributing to the overall improvement of the quality of the manuscript.

Reviewer: 1

Reviewer Comment:

The authors have adequately addressed my comments and provided substantial edits in the revised manuscript. Together with the additional clarifications that were provided for the other two reviewers, the current manuscript is more transparent and argumentation on the promising new method justifiable. I do not have any further comments.

Authors' response:

We thank the reviewer for investing time into providing us with valuable feedback for our work.

Reviewer: 2

Reviewer Comment:

All my concerns are addressed in the revised version of the article, and I also believe the results are promising and impactful. I do not have any further comments.

Authors' response:

We thank the reviewer for investing time into providing us with valuable feedback for our work.